# Plasma Galectin-3 predicts deleterious vascular dysfunction affecting post-myocardial infarction patients: An explanatory study

Olivier Huttin[1,2], Damien Mandry[3,4], Batric Popovic[1,2], Patrick Rossignol[1,5,6], Freddy Odille[3,5], Emilien Micard[3,5], Zohra Lamiral[5], Faïez Zannad[1,2,5], Nicolas Girerd[1,2,5], Pierre-Yves Marie[1,7] *

1 Université de Lorraine, INSERM, UMR-1116, Nancy, France, 2 Department of Cardiology, CHRU-Nancy, Université de Lorraine, Nancy, France, 3 Université de Lorraine, INSERM, UMR-1254, Nancy, France, 4 Department of Radiology, CHRU-Nancy, Université de Lorraine, Nancy, France, 5 Université de Lorraine, CHRU-Nancy, INSERM, CIC 1433, Nancy, France, 6 FCRIN INI-CRCT, Nancy, France, 7 CHRU-Nancy, Université de Lorraine, Nuclear Medicine & Nancyclotep Imaging Platform, Nancy, France

* py.marie@chru-nancy.fr

**Data Availability Statement:** All relevant data are within the paper and Supporting Information files.

**Funding:** The funders had no role in study design, data collection and analysis, decision to publish, or

## Abstract

### Objectives

In a previous analysis of a post-myocardial infarction (MI) cohort, abnormally high systemic vascular resistances (SVR) were shown to be frequently revealed by MRI during the healing period, independently of MI severity, giving evidence of vascular dysfunction and limiting further recovery of cardiac function. The present ancillary and exploratory analysis of the same cohort was aimed at characterizing those patients suffering from high SVR remotely from MI with a large a panel of cardiovascular MRI parameters and blood biomarkers.

### Methods

MRI and blood sampling were performed 2–4 days after a reperfused MI and 6 months thereafter in 121 patients. SVR were monitored with a phase-contrast MRI sequence and patients with abnormally high SVR at 6-months were characterized through MRI parameters and blood biomarkers, including Galectin-3, an indicator of cardiovascular inflammation and fibrosis after MI. SVR were normal at 6-months in 90 patients (SVR-) and abnormally high in 31 among whom 21 already had high SVR at the acute phase (SVR++) while 10 did not (SVR+).

### Results

When compared with SVR-, both SVR+ and SVR++ exhibited lower recovery in cardiac function from baseline to 6-months, while baseline levels of Galectin-3 were significantly different in both SVR+ (median: 14.4 (interquartile range: 12.3–16.7) ng.mL$^{-1}$) and SVR++ (13.0 (11.7–19.4) ng.mL$^{-1}$) compared to SVR- (11.7 (9.8–13.5) ng.mL$^{-1}$, both p < 0.05). Plasma Galectin-3 was an independent baseline predictor of high SVR at 6-months (p =

preparation of the manuscript. Academic fundings were obtained through the French "Programme Hospitalier de Recherche Clinique" (2010) and the RHU FIGHT-HF program (ANR-15-RHU-0004)

**Competing interests:** The authors have declared that no competing interests exist.

0.002), together with the baseline levels of SVR and left ventricular end-diastolic volume, whereas indices of MI severity and left ventricular function were not. In conclusion, plasma Galectin-3 predicts a deleterious vascular dysfunction affecting post-MI patients, an observation that could lead to consider new therapeutic targets if confirmed through dedicated prospective studies.

## Introduction

In the previous "REMI" (relation between aldosterone and cardiac REmodeling after Myocardial Infarction) cohort, a lower recovery in cardiac function was documented in patients for whom systemic vascular resistances (SVR) were abnormally high during the post-myocardial infarction (MI) healing period, independently of MI severity and in spite of the commonly prescribed vasodilator regimens (Angiotensin Converting Enzyme Inhibitors (ACEI) or Angiotensin Receptor Blockers (ARBs)) [1]. Such patients with high SVR may be difficult to detect after MI, as well as in the more general setting of heart failure, hypertension being frequently masked by decreases in cardiac contractility and stroke volume [1,2]. In these situations, it is likely that SVR measurements by non-invasive techniques [1–4] may help in assessing the usefulness of further decreasing SVR by vasodilating treatments. Such decreases in SVR were indeed previously shown to provide proportional enhancements in cardiac output after MI [5].

Furthermore, the mechanism of this vascular dysfunction, leading to high SVR in spite of post-MI vasodilator treatment, warrants further clarification. It is likely that the renin-angiotensin-aldosterone system (RASS), a key modulator of vascular function and ischemic remodeling, should be assessed in this setting [6,7], as well as certain biomarkers of inflammation and fibrosis. This is particularly the case of Galactin-3, a plasma biomarker of cardiovascular inflammation and fibrosis [8], which is an established predictor of cardiac remodeling and outcome of post-MI patients and which was recently shown to be linked to SVR in certain populations with inflammatory diseases [9]. This analysis should also consider certain hemodynamic factors, especially the fact that higher SVR are required for maintaining a sufficiently high blood pressure (BP) in patients presenting the lowest stroke volumes [3,5,10,11].

In light of the above, this ancillary and exploratory analysis of the "REMI" post-MI cohort [1] was aimed at characterizing those patients suffering from high SVR remotely from MI with a large a panel of cardiovascular MRI parameters and blood biomarkers.

## Material and methods

### Study population

As previously described in detail for this "REMI" (relation between aldosterone and cardiac REmodeling after Myocardial Infarction) cohort [1], patients successfully treated by primary percutaneous transluminal coronary angioplasty for a first MI and with an initial occlusion or sub-occlusion of the MI-related coronary artery at angiography, were prospectively included. Main exclusion criteria were: any other significant cardiac disease, any contraindication to MRI, absence of sinus cardiac rhythm, a multivessel disease at coronary angiography, and a >12h delay-time between the onset of chest pain and reperfusion.

All subjects gave signed informed consent to participate. The study protocol complied with the principles of the Declaration of Helsinki, was approved by the local Ethics Committee

(Comité de Protection des Personnes EST-III, agreement n° 2009-A00537-50) and registered on the ClinicalTrials.gov site (NCT01109225). The protocol of the REMI study is available as S2 Protocol.

### Study design

Blood sampling and cardiovascular MRI were performed at 2 to 4 days after acute MI reperfusion and 6 months (± 15 days) later. Patients showing abnormally high SVR at 6 months were compared with the other study patients for MRI parameters of cardiac and vascular function and of infarct size, as well as for plasma biomarkers of heart failure (Brain Natriuretic Peptide (BNP)), myocardial necrosis (peak Creatine Kinase-MB and Troponin) and systemic inflammation and/or RASS activation (C-Reactive Protein, Neutrophil Gelatinase-Associated Lipocalin (NGAL [12]), Galectin-3 [13], active Renin and Aldosterone [6]. Glomerular filtration rate (eGFR) was estimated in ml/min per 1.73 $m^2$ body surface area with the CKD-EPI equation [14].

### Cardiovascular MRI

As detailed previously [1,3,4], MRI exams were performed on a single 3.0 Tesla magnet (Signal HDxt, GE Healthcare, Milwaukee, Wisconsin) with a cardiac coil. Systolic, diastolic and mean brachial blood BP were measured with an automated sphygmomanometer (Maglife C, Schiller Medical, Wissembourg, France). Three measurements were obtained during each MRI examination and mean values were stored for analyses herein.

A steady-state free precession pulse sequence was used to assess cardiac function in contiguous short axis planes, as previously detailed [3,4], and LV end-diastolic volume, end-systolic volume, LV mass and ejection fraction were obtained using dedicated software (MASS research v2013-exp™, Medis, Leiden University Medical Center, The Netherlands). The LV concentric remodeling index was computed as the LV mass over end-diastolic volume ratio [1,3,4].

The MI area was analyzed on 8 to 10 short axis slices covering the LV volume and on vertical and horizontal mid-ventricular long-axis slices, which were all recorded with a T1-weighted segmented phase-sensitive inversion-recovery (PSIR) sequence, 10 to 15 minutes after the injection of a gadolinium-labeled tracer (0.1 mmol.$kg^{-1}$ body weight of Dotarem®, GUERBET, France). The MI volume was considered as that showing a late gadolinium enhancement by visual analysis and was expressed in % of the total LV volume by using a 17-segment LV division and while taking into account the number of quartiles involved in each segment [1]. The MI volume with microvascular obstruction was determined as that showing a central hypo-enhancement within the bright signal of delayed enhancement [1].

Aortic stroke volume (SV) indexed to body surface area was determined in the ascending aorta by using a velocity-encoded phase-contrast gradient-echo sequence and the "CV flow" quantification software (Leiden University Medical Center, Medis, The Netherlands) [1,3,4]. Indexed SV was used to calculate cardiac index (SV x heart rate) and systemic vascular resistance (SVR: mean BP / cardiac index). SVR values above 40 mmHg.min.$m^2$.$L^{-1}$ were considered as abnormal. This threshold corresponds to the upper limit of the 95% confidence interval in an already-described normal population of 100 subjects with comparable age range and MRI protocol as that in the population of subjects in the current study [3].

### Statistical analyses

All analyses were performed using the SAS software version 9.4 (SAS Institute Inc., Cary, NC, USA). The two-tailed significance level was set at P <0.05.

Continuous variables are expressed as median with interquartile range (Q1 –Q3) and categorical variables as frequencies (percentages). Comparisons of characteristics between SVR groups were carried out using non parametric Kruskall-Wallis tests for continuous variables and Fisher's exact tests for categorical variables.

Associations between the baseline characteristics and the SVR status were additionally assessed using univariable and multivariable ordinal logistic regression models with baseline characteristics as explanatory variables and the three SVR categories as outcome variable, namely SVR- (as reference category), SVR+ and SVR++. Odd ratios (ORs) are reported with 95% confidence interval. Overall p-values of univariable logistic regressions were corrected for multiple testing using a false discovery rate (FDR) of 5%, applying the Benjamini–Hochberg procedure.

Assumption of log linearity for continuous variables was verified using restricted cubic spline with 3 knots. When log linearity was not met, variables were dichotomized according to the median.

A multivariable ordinal logistic regression model was built using a backward selection procedure (p-to-remove = 0.10 because of the small sample size) applied to variables with corrected overall p-values less than 0.20 at univariable logistic regression analyses.

## Results

### Baseline patient characteristics

A total of 141 patients were initially included. However, MRI was not performed at 6 months in 3 patients due to contraindications and another 17 due to consent withdrawals, thereby leaving 121 patients for the final analysis (see flowchart in Fig 1). Median age was 57.7 years (interquartile range: 49.0–63.2 years), 18 (15%) were women and the MI-related vessel was the left anterior descending artery in 63 patients (52%).

### Six-month evolution

At 6 months, 101 of the 121 patients (84%) were under beta-blocker treatment and 104 (86%) were under ACEI although only half received the recommended prescribed dose targeted in post-MI trials. Only 6 patients (5%) received mineralocorticoid receptor antagonist treatment.

In the overall population, there were significant improvement in cardiac function with increases from baseline to 6 months in LV ejection fraction (42.8% (37.6%-48.2%) vs. 49.9% (42.8%-54.8%), $p < 0.001$) and in cardiac index (in $L.min^{-1}.m^{-2}$: 2.40 (2.15–2.71) vs. 2.58 (2.37–2.97), $p < 0.001$), together with a significant decrease in SVR (in $mmHg.min.m^2.L^{-1}$: 39.4 (31.7–44.0) vs. 34.8 (29.8–40.2), $p = 0.001$). SVR were normal at 6 months in 90 patients (SVR- group), but abnormally high in 31 (26%), among whom 21 already had high SVR at baseline (SVR++ group) while the remaining 10 did not (SVR+ group).

As detailed in Table 1, the SVR-, SVR+ and SVR++ groups were comparable in terms of medical regimen with high rates for beta-blockers and ACEI in all groups. However, when compared with SVR-, both SVR+ and SVR++ had evidence of a lower cardiac recovery from baseline to 6 months with a lower increase in LV ejection fraction (for the difference between 6 months and baseline, SVR+: +2.7% (-2.0% - 5.3%) and SVR++: +4.5% (0.8% - 8.4%) vs. SVR-: +8.7% (4.0% - 12.8%), both $p \leq 0.05$) and with a lower cardiac index achieved at 6 months (in $L.min^{-1}.m^{-2}$: SVR+: 2.12 (1.94–2.38) and SVR++: 2.14 (1.92–2.47) vs. SVR-: 2.70 (2.47–3.12), both $p < 0.05$) (Table 2).

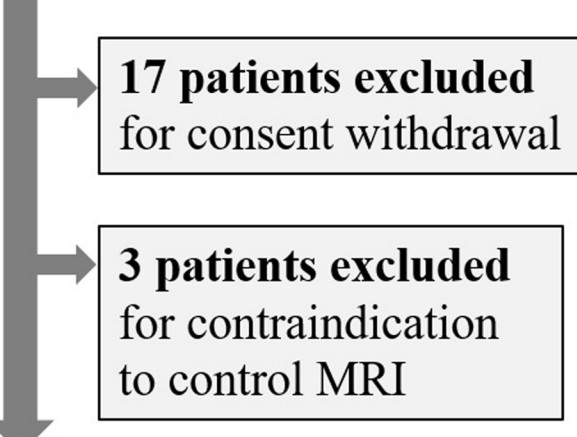

**141 patients prospectively included** and who had blood sampling and MRI at 2 to 4 days after acute MI reperfusion

**17 patients excluded** for consent withdrawal

**3 patients excluded** for contraindication to control MRI

**121 patients had the 6-month control** with additional blood sampling and MRI.

**Fig 1. Flowchart.**

### Baseline predictors of abnormally high SVR at 6 months

Although patient distribution between the SVR-, SVR+ and SVR++ groups was significantly related to several baseline variables (see Tables 1 and 2), only the baseline level of Galectin-3 was significantly different in both SVR+ (14.4 (12.3–16.7) ng.mL$^{-1}$) and SVR++ (13.0 (11.7–19.4) ng.mL$^{-1}$) compared to SVR- (11.7 (9.8–13.5) ng.mL$^{-1}$, both $p < 0.05$).

In addition, patients from the SVR++ group had a high rate of history of hypertension (52%, Table 1) and a particular hemodynamic profile at baseline involving not only higher SVR, but also higher LV concentric remodeling, smaller LV end-diastolic volumes and smaller stroke volumes (Table 2).

By contrast, the SVR-, SVR+ and SVR++ groups were comparable for all baseline indices of MI severity and cardiac function–i.e. Creatine Kinase-MB and Troponin peaks, BNP level, MRI areas of MI or microvascular obstruction, and LV ejection fraction (Tables 1 and 2).

Finally, in ordinal logistic regression analyses (Table 3), univariable baseline predictors of SVR+ and SVR++ patients were: plasma Galectin-3, a LV end-diastolic volume < 90 mL.m$^{-2}$, a diastolic blood pressure > 70 mmHg, and all SV-derived parameters (SV, cardiac index, SVR). The three best independent baseline predictors selected through multivariable analysis were plasma Galectin-3 (p = 0.010), a LV end-diastolic volume < 90 mL.m$^{-2}$ (p = 0.022) and SVR (p = 0.003) (Table 3). These 3 selected parameters remained unchanged when age and/or renal function (estimated by the glomerular filtration rate) were forced into the model.

**Table 1. Comparisons of clinical variables and blood biomarkers between patients with normal SVR at 6 months (SVR-), those with abnormal SVR only at 6 months (SVR+) and those with abnormal SVR both at baseline and 6 months (SVR++).**

| | | Baseline variables | | | | 6-month variables | | |
|---|---|---|---|---|---|---|---|---|
| | SVR- (n = 90) | SVR+ (n = 10) | SVR++ (n = 21) | P value | SVR- (n = 90) | SVR+ (n = 10) | SVR++ (n = 21) | P value |
| Age (years) | 56 (49–61) | 56 (47–63) | 58 (49–72) | 0.54 | _____ | _____ | _____ | _____ |
| Female gender | 13 (14%) | 2 (20%) | 3 (14%) | 0.91 | _____ | _____ | _____ | _____ |
| History of hypertension | 25 (28%) | 3 (30%) | 11 (52%) | 0.10 | _____ | _____ | _____ | _____ |
| Body mass index (kg.m$^{-2}$) | 24 (22–28) | 26 (24–28) | 25 (23–28) | 0.54 | 24 (22–27) | 26.9 (24–28) | 25 (24–28) | 0.15 |
| ACEI treatment | 77 (85%) | 8 (80%) | 18 (86%) | 0.91 | 78 (87%) | 10 (100%) | 16 (76%) | 0.22 |
| Beta-blocker treatment | 73 (81%) | 9 (90%) | 18 (86%) | 0.85 | 75 (83%) | 10 (100%) | 16 (76%) | 0.31 |
| Peak Creatine Kinase-MB (U.L$^{-1}$) | 2513 (1052–4000) | 3647 (1374–4678) | 2209 (1263–2984) | 0.40 | _____ | _____ | _____ | _____ |
| Ln (peak Troponin (µg.L$^{-1}$)) | 4.6 (4.3–6.3) | 5.3 (4.5–7.9) | 4.6 (4.1–5.2) | 0.58 | _____ | _____ | _____ | _____ |
| eGFR (mL.min$^{-1}$ and per 1.73 m$^{-2}$ of BSA) | 83.0 (72–94) | 77.0 (70–83) | 73.0 (65–82) | 0.10 | _____ | _____ | _____ | _____ |
| C-Reactive Protein (mg.L$^{-1}$) | 12.7 (5.5–22.8) | 21.2 (13.7–52.2) | 12.9 (10.2–26.4) | 0.25 | 1.1 (0.6–2.1) | 2.0 (0.6–2.2) | 1.1 (0.6–2.9) | 0.43 |
| Brain Natriuretic Peptide (pg.mL$^{-1}$) | 157 (82–266) | 177 (130–300) | 113 (62–205) | 0.19 | 52 (25–119) | 52 (23–77) | 63 (25–129) | 0.79 |
| NGAL (ng.mL$^{-1}$) | 70 (59–87) | 84 (61–107) | 75 (65–107) | 0.14 | 64 (54–84) | 88 (57–115) | 79 (73–93)* | 0.014 |
| Renin (pg.mL$^{-1}$) | 14.4 (6.8–28.2) | 18.9 (5.4–201.1) | 20.6 (9.0–80.0) | 0.25 | 26.9 (12.6–65.9) | 40.9 (13.8–128.0) | 33.6 (10.9–91.8) | 0.71 |
| Aldosterone (pg.mL$^{-1}$) | 22.8 (15.6–31.0) | 26.3 (20.5–72.4) | 23.1 (20.1–40.0) | 0.29 | 58.3 (37.6–97.5) | 67.5 (38.9–196.9) | 64.2 (41.7–115.9) | 0.41 |
| Galectin-3 (ng.mL$^{-1}$) | 11.7 (9.8–13.5) | 14.4 (12.3–16.7)* | 13.0 (11.7–19.4)* | 0.003 | 11.9 (10.2–13.2) | 12.3 (10.9–15.4) | 12.1 (10.8–16.5) | 0.29 |

ACEI, Angiotensin Converting Enzyme Inhibitors; BSA: body surface area; eGFR, estimated glomerular filtration rate; NGAL, Neutrophil Gelatinase-Associated Lipocalin.

*: p<0.05 for comparisons of SVR+ or SVR++ vs. SVR-, and

†: p<0.05 for comparisons of SVR++ vs. SVR+ at the same time point (baseline or 6 months).

## Discussion

The main findings of the present study are that patients at risk of suffering from high SVR after 6 months of post-MI treatment: 1) did not have higher MI severity or LV dysfunction at baseline, as assessed by cardiac enzymes and MRI variables, and 2) may be predicted at baseline by Galactin-3 plasma level.

High SVR are mainly due to structural and functional changes in small arteries with a constant decrease in lumen diameter and possible increases in wall thickness and wall fibrosis. All of these structural changes are commonly documented during the normal aging process, together with an increase in the stiffness of large arteries, and may be accelerated by hypertension and various metabolic and inflammatory disorders [15]. However, SVR may also increase in response to various neurohormonal factors such as those triggered for increasing perfusion pressures in hypovolemic shock or heart failure [5].

A vascular dysfunction, leading to high SVR, was already shown to be potentially deleterious in post-MI patients, due to increased LV wall stress [5,16] and, as observed in the present study cohort, likely limiting subsequent recovery in LV ejection fraction and cardiac output. An at-least partial recovery of cardiac function is a common observation during the post-MI healing period [1] and is also a consequence of the prescribed vasodilator therapies with the increase in stroke volume and cardiac output being proportional to the decrease in SVR [5].

**Table 2. Comparisons of hemodynamic and cardiovascular MRI variables between patients with normal SVR at 6 months (SVR-), those with abnormal SVR only at 6 months (SVR+) and those with abnormal SVR both at baseline and 6 months (SVR++).**

| | | Baseline variables | | | | 6-month variables | | |
|---|---|---|---|---|---|---|---|---|
| | SVR- (n = 90) | SVR+ (n = 10) | SVR++ (n = 21) | P value | SVR- (n = 90) | SVR+ (n = 10) | SVR++ (n = 21) | P value |
| Heart rate (bpm) | 64 (57–72) | 67 (61–73) | 62 (59–71) | 0.70 | 58 (52–62) | 55 (53–57) | 55 (53–61) | 0.38 |
| Systolic blood pressure (mmHg) | 125 (110–139) | 111 (101–126) | 138 (123–149)[†] | 0.019 | 124 (115–134) | 129 (122–143) | 150 (140–163)* | < 0.001 |
| Diastolic blood pressure (mmHg) | 73 (65–82) | 65 (61–70) | 79 (73–85)[†] | 0.005 | 68 (63–75) | 76 (66–84) | 80 (72–90)* | < 0.001 |
| Mean blood pressure (mmHg) | 90 (81–102) | 83 (75–89) | 99 (91–105)[†] | 0.005 | 86 (80–95) | 93 (84–105) | 104 (95–114)* | < 0.001 |
| LV ejection fraction (%) | 43 (37–48) | 42 (33–49) | 43 (39–47) | 0.91 | 51 (45–56) | 42 (37–50)* | 49 (44–52) | 0.034 |
| LV end-diastolic volume (mL.m$^{-2}$) | 94 (83–105) | 92 (86–104) | 82 (69–87)*[†] | < 0.001 | 99 (85–107) | 88 (83–100) | 87 (75–92)* | 0.005 |
| LV end-systolic volume (mL.m$^{-2}$) | 53 (44–63) | 54 (46–63) | 45 (39–51)* | 0.004 | 47 (39–58) | 50 (43–61) | 43 (35–50) | 0.21 |
| LV mass (g.m$^{-2}$) | 55 (49–60) | 53 (49–58) | 52 (47–56) | 0.39 | 46 (42–52) | 43 (39–48) | 46 (41–49) | 0.46 |
| LV concentric remodeling index | 0.58 (0.53–0.64) | 0.56 (0.54–0.58) | 0.63 (0.58–0.73)* | 0.029 | 0.47 (0.42–0.53) | 0.49 (0.41–0.53) | 0.56 (0.52–0.59)* | 0.003 |
| MI volume at MRI (% of LV) | 21 (12–31) | 26 (7–34) | 22 (13–28) | 0.87 | 13 (7–24) | 20 (7–28) | 12 (10–19) | 0.66 |
| Microvascular obstruction (% of LV) | 2 (0–9) | 4 (0–12) | 0 (0–9) | 0.82 | —— | —— | —— | —— |
| Stroke volume index (mL.m$^{-2}$) | 39.7 (34.0–43.9) | 38 (32–43) | 33.8 (28.2–37.0)* | 0.003 | 48.6 (44.8–53.9) | 42.4 (34.3–45.2)* | 38.6 (32.1–45.4)* | < 0.001 |
| Cardiac index (L.min$^{-1}$.m$^{-2}$) | 2.52 (2.18–2.82) | 2.41 (2.31–2.67) | 2.22 (1.96–2.31)*[†] | 0.002 | 2.70 (2.47–3.12) | 2.12 (1.94–2.38)* | 2.14 (1.92–2.47)* | < 0.001 |
| SVR (mmHg.min.m$^2$.L$^{-1}$) | 36.7 (31.1–42.8) | 34.1 (29.8–35.8) | 44.5 (42.9–47.7)*[†] | <0.001 | 31.8 (28.2–35.4) | 42.4 (40.7–48.7)* | 48.5 (43.9–52.1)* | <0.001 |

LV, left ventricle; MI, myocardial infarction; NS, non significant with a p value < 0.10; SVR, systemic vascular resistance

*: p<0.05 for comparisons of SVR+ or SVR++ vs. SVR-, and

[†]: p<0.05 for comparisons of SVR++ vs. SVR+ at the same time point (baseline or 6 months).

The present observation of a strong and independent relationship between elevated plasma Galectin-3 and increasing SVR was also documented recently, but in a very different cohort of patients with low-grade systemic inflammation (long-standing rheumatoid arthritis) [9]. This relationship is further strengthened by prior observations that plasma Galectin-3 is independently related to various systemic arterial diseases in asymptomatic individuals (aortic stiffness, atherosclerosis) [9,17,18].

Galectin-3 is mainly expressed in fibroblasts, endothelial cells as well as in inflammatory cells such as activated macrophages, and is considered a key link between inflammation and fibrosis for the cardiovascular system [8]. While Galectin-3 is a profibrotic agent in itself, it also mediates aldosterone-induced fibrosis within the vessels, as well as within the heart and kidney [13]. It additionally regulates chronic vascular inflammation, promoting osteogenic differentiation of vascular smooth muscle cells and vessel calcification [19], and furthermore acts as an amplifier of inflammation in atherosclerotic plaque progression through macrophage activation and monocyte chemoattraction [20].

In clinical routine, Galectin-3 can be used to improve risk stratification in heart failure patients [21], and its predictive value for adverse cardiovascular events has also been demonstrated in non-heart failure patients [22].

**Table 3. Univariable and multivariable baseline predictors of SVR+ and SVR++ patients with odds ratios (OR) and 95% confidence intervals.**

| Variable | Response | Univariable ordinal regression | | Multivariable ordinal regression | |
|---|---|---|---|---|---|
| | | OR (95% CI) | Overall P-value | OR (95%CI) | Overall P-value |
| Age (per 10-year increment) | SVR+ | 0.899 (0.466, 1.734) | 0.56 | | |
| | SVR++ | 1.410 (0.892, 2.231) | | | |
| Female gender | SVR+ | 1.481 (0.282, 7.766) | 0.90 | | |
| | SVR++ | 0.987 (0.254, 3.832) | | | |
| History of hypertension | SVR+ | 1.114 (0.267, 4.652) | 0.25 | | |
| | SVR++ | 2.860 (1.081, 7.565) | | | |
| Body mass index (per 5 kg.m-2 increment) | SVR+ | 1.341 (0.616, 2.917) | 0.61 | | |
| | SVR++ | 1.445 (0.818, 2.554) | | | |
| ACEI treatment | SVR+ | 0.675 (0.129, 3.542) | 0.90 | | |
| | SVR++ | 1.013 (0.261, 3.932) | | | |
| Beta-blocker treatment | SVR+ | 2.096 (0.248, 17.677) | 0.90 | | |
| | SVR++ | 1.397 (0.369, 5.290) | | | |
| Peak Creatine Kinase-MB (per 1000 U.L$^{-1}$) | SVR+ | 1.195 (0.840, 1.699) | 0.63 | | |
| | SVR++ | 0.899 (0.672, 1.202) | | | |
| Peak Troponin (µg.L$^{-1}$) | SVR+ | 1.277 (0.896, 1.821) | 0.61 | | |
| | SVR++ | 0.911 (0.677, 1.226) | | | |
| eGFR (per 10 mL.min$^{-1}$ increment) | SVR+ | 0.893 (0.589, 1.353) | 0.27 | | |
| | SVR++ | 0.724 (0.537, 0.976) | | | |
| C-Reactive Protein (mg.L$^{-1}$) | SVR+ | 1.151 (0.998, 1.327) | 0.36 | | |
| | SVR++ | 1.011 (0.864, 1.182) | | | |
| Brain Natriuretic Peptide (per 10 pg.mL$^{-1}$ increment) | SVR+ | 1.021 (0.987, 1.055) | 0.42 | | |
| | SVR++ | 0.976 (0.938, 1.017) | | | |
| NGAL (per 10 ng.mL$^{-1}$ increment) | SVR+ | 1.236 (1.012, 1.511) | 0.20 | | |
| | SVR++ | 1.158 (0.985, 1.361) | | | |
| Renin >15 pg.mL$^{-1}$ | SVR+ | 1.635 (0.948, 2.821) | 0.75 | | |
| | SVR++ | 1.499 (0.971, 2.313) | | | |
| Aldosterone (pg.mL$^{-1}$) | SVR+ | 3.120 (0.627, 15.531) | 0.90 | | |
| | SVR++ | 2.496 (0.841, 7.409) | | | |
| Galectin-3 (ng.mL$^{-1}$) | SVR+ | 1.204 (1.040, 1.394) | 0.048 | 1.213 (1.040, 1.414) | 0.010 |
| | SVR++ | 1.186 (1.057, 1.331) | | 1.163 (1.023, 1.323) | |
| Heart rate (per10 bpm increment) | SVR+ | 1.145 (0.645, 2.032) | 0.90 | | |
| | SVR++ | 1.022 (0.663, 1.575) | | | |
| Systolic blood pressure >130 mmHg | SVR+ | 0.412 (0.083, 2.054) | 0.20 | | |
| | SVR++ | 2.676 (1.006, 7.120) | | | |
| Diastolic blood pressure >70 mmHg | SVR+ | 0.159 (0.032, 0.793) | 0.048 | | |
| | SVR++ | 3.818 (1.047, 13.921) | | | |
| Mean blood pressure >93 mmHg | SVR+ | 0.342 (0.069, 1.703) | 0.36 | | |
| | SVR++ | 1.825 (0.698, 4.766) | | | |
| LV ejection fraction (per 5% increment) | SVR+ | 0.969 (0.642, 1.462) | 0.90 | | |
| | SVR++ | 1.091 (0.796, 1.494) | | | |
| LV end-diastolic volume < 90 mL.m$^{-2}$ | SVR+ | 1.048 (0.276, 3.978) | 0.048 | 1.227 (0.290, 5.208) | 0.022 |
| | SVR++ | 6.679 (2.075, 21.491) | | 6.666 (1.733, 25.641) | |
| LV end-systolic volume < 50 mL.m$^{-2}$ | SVR+ | 0.833 (0.220, 3.156) | 0.36 | | |
| | SVR++ | 2.500 (0.922, 6.782) | | | |
| LV mass (per 10 g.m$^{-2}$) | SVR+ | 1.055 (0.532, 2.092) | 0.70 | | |
| | SVR++ | 0.731 (0.431, 1.238) | | | |

*(Continued)*

**Table 3.** (Continued)

| Variable | Response | Univariable ordinal regression | | Multivariable ordinal regression | |
|---|---|---|---|---|---|
| | | OR (95% CI) | Overall P-value | OR (95%CI) | Overall P-value |
| LV concentric remodeling index>0.58 | SVR+ | 0.335 (0.066, 1.700) | 0.17 | | |
| | SVR++ | 2.927 (1.040, 8.235) | | | |
| MI volume at MRI (% of LV) | SVR+ | 1.189 (0.678, 2.084) | 0.90 | | |
| | SVR++ | 0.951 (0.632, 1.432) | | | |
| Microvascular obstruction | SVR+ | 1.374 (0.722, 2.616) | 0.82 | | |
| (per 10% of LV increment) | SVR++ | 1.044 (0.600, 1.819) | | | |
| Stroke volume index >38 mL.m$^{-2}$ | SVR+ | 0.765 (0.207, 2.828) | 0.048 | | |
| | SVR++ | 0.127 (0.035, 0.464) | | | |
| Cardiac index >2.4 L.min-1.m$^{-2}$ | SVR+ | 0.731 (0.198, 2.704) | 0.06 | | |
| | SVR++ | 0.172 (0.054, 0.552) | | | |
| SVR (mmHg.min.m$^2$.L$^{-1}$) | SVR+ | 0.914 (0.824, 1.013) | 0.03 | 0.905 (0.812, 1.008) | 0.003 |
| | SVR++ | 1.123 (1.044, 1.208) | | 1.109 (1.031, 1.194) | |

eGFR: estimated glomerular filtration rate; LV, left ventricle; MI: myocardial infarction; NGAL, Neutrophil Gelatinase-Associated Lipocalin; SVR, systemic vascular resistances.

Previous studies have failed to clearly and consensually show that Galectin-3 was a predictor of ventricular remodeling after MI [20], and in the present study, the plasma level of Galectin-3 was not directly predictive of this remodeling—i.e. with a lack of any significant correlation between baseline Galectin-3 and the 6-month evolutions in LV volume or EF (results not shown). This is presumably due to the complex cardiac effects of Galectin-3 with, on the one hand, an upregulation that is known to be highly beneficial in the initial phases of tissue repair and, on the other, an overexpression that could be associated with prolonged inflammation and adverse remodeling [20].

This predictive value is seemingly higher for the post-MI remodeling of systemic arteries since, in our multivariable analysis, a higher Galectin-3 was one of the independent predictors of high SVR at 6 months, with a smaller LV end-diastolic volume and a higher baseline SVR being the other predictors. Smaller end-diastolic volumes are commonly associated with lower cardiac output and thus with higher SVR for maintaining a sufficiently high BP [3,5,10,11]. This hemodynamic profile has been previously associated with hypertensive or pre-hypertensive states [3,23], as well as with low exercise training and/or low exercise capacity [10,11].

However, this particular hemodynamic profile was definitely absent at baseline in one third of our patients who nevertheless featured high SVR at 6 months (i.e. those from our SVR + group). This smaller subgroup had different, albeit less precise, baseline characteristics with trends toward lower BP and SVR levels, as well as higher plasma CRP, when compared to the two other groups (Tables 1 and 2). This suggests a distinct inflammatory and hemodynamic context, warranting further analyses in larger populations.

Nevertheless, as already discussed above, these SVR+ patients, as well as the SVR++ patients, were characterized by a higher plasma Galectin-3 level at baseline. In experimental models, anti-aldosterone drugs were already shown to reverse the vascular fibrosis induced by Galectin-3 [13]. Therefore, it could be wondered whether an enhancement in the pharmacological blockade of the mineralocorticoid pathway and/or of the RAAS might be beneficial if prescribed in a highly selected population of patients for whom plasma Galectin-3 is particularly high at baseline. This is all the more true given that half of our patients did not receive, at 6 months, the doses of ACEI or ARBs targeted in post-MI trials. Such sub-optimal medical

regimens have already been documented and explained by contraindications and intolerance issues but also by an underestimation of treatment benefit [24–26].

Finally, several carbohydrate- or peptide-based inhibitors of Galectin-3 are under development, particularly for oncologic indications [27]. In the future, their effects on post-MI patients and especially on those with high plasma Galectin-3 levels could likely be the subject of dedicated studies.

Except for physical activity, the main factors known to drive SVR were tested in the present study–i.e. age, kidney function, cardiac function, drug treatment (in particular ACEI/ARBs and beta-blockers)–none of which were found to provide any additional predictive information with regard to that achieved by the baseline levels of Galectin-3, LV end-diastolic volume and SVR. This could at least be partly explained by the low sample size of the present study population. However, this may also be explained by the particular conditions of the post-MI period where SVR are strongly affected by a cascade of adaptive hemodynamic mechanisms and neurohormonal changes and by the introduction of interfering vasoactive treatments, thereby minimizing the impact of other factors.

The main limitation of the present study is its exploratory nature and thus, further dedicated, larger-scale prospective studies, designed at providing a more accurate assessment of the predictive value of Galectin-3 in this setting, are required.

## Conclusion

This ancillary and observational analysis of the "REMI" cohort shows firstly that patients suffering from high SVR remotely from MI exhibit a lower recovery of cardiac function and secondly, that the risk of such vascular dysfunction may be predicted by higher plasma Galectin-3, but not by indices of MI severity or left ventricular function. Although these finding need to be confirmed through further dedicated prospective studies, the present observational results suggest particular mechanisms and potential therapeutic targets for further decreasing SVR and thereby enhancing cardiac function in selected patient groups.

## Supporting information

**S1 Dataset.**
(XLS)

**S1 Checklist. CONSORT 2010 checklist of information to include when reporting a randomised trial**\*.
(DOC)

**S1 Protocol.**
(DOC)

**S2 Protocol.**
(DOCX)

## Author Contributions

**Conceptualization:** Olivier Huttin, Faïez Zannad, Pierre-Yves Marie.

**Formal analysis:** Zohra Lamiral, Nicolas Girerd.

**Funding acquisition:** Patrick Rossignol.

**Investigation:** Olivier Huttin, Damien Mandry, Batric Popovic, Pierre-Yves Marie.

**Methodology:** Damien Mandry, Freddy Odille, Emilien Micard, Zohra Lamiral.

**Resources:** Olivier Huttin, Damien Mandry, Batric Popovic, Freddy Odille, Emilien Micard, Faïez Zannad, Nicolas Girerd, Pierre-Yves Marie.

**Software:** Emilien Micard.

**Supervision:** Patrick Rossignol, Faïez Zannad.

**Validation:** Damien Mandry, Patrick Rossignol, Zohra Lamiral, Faïez Zannad, Nicolas Girerd.

**Writing – original draft:** Pierre-Yves Marie.

**Writing – review & editing:** Olivier Huttin, Damien Mandry, Patrick Rossignol, Faïez Zannad, Nicolas Girerd, Pierre-Yves Marie.

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
