## [Decision Letter · Decision Letter 0]

8 Jan 2020

PONE-D-19-29784

Plasma Galectin-3 predicts deleterious vascular dysfunction affecting post-myocardial infarction patients

PLOS ONE

Dear Prof. Marie,

Thank you for submitting your manuscript to PLOS ONE. After careful consideration, we feel that it has merit but does not fully meet PLOS ONE’s publication criteria as it currently stands. Therefore, we invite you to submit a revised version of the manuscript that addresses the points raised during the review process.

Reviewer 1 has raised concerns about the actual value of the models. Indeed, they underscore the established role of other variables associated with SVR. In addition, they suggest a more comprehensive review of the existing literature. Reviewer 2 har raised specific comments with respect to statistical analysis.   

We would appreciate receiving your revised manuscript by Feb 22 2020 11:59PM. To enhance the reproducibility of your results, we recommend that if applicable you deposit your laboratory protocols in protocols.io, where a protocol can be assigned its own identifier (DOI) such that it can be cited independently in the future. For instructions see: http://journals.plos.org/plosone/s/submission-guidelines#loc-laboratory-protocols

We look forward to receiving your revised manuscript.

Kind regards,

Giuseppe Andò, M.D., Ph.D.

Academic Editor

PLOS ONE

Journal Requirements:

2. Thank you for including your ethics statement: All subjects gave signed informed consent to participate. The study protocol complied with the principles of the Declaration of Helsinki, was approved by the local Ethics Committee (CPP agreement n° 2009-A00537-50) and registered on the ClinicalTrials.gov site (NCT01109225).

4. Thank you for including your dual publication statement.  To comply with PLOS ONE submission guidelines can you please upload a copy of the previous publication as a supplementary file.  Please be aware that if we locate a dual publication issue we will require copyright holder’s permission to publish under CC-By 4.0.

Reviewers' comments:

Reviewer's Responses to Questions

**Comments to the Author**

1. Is the manuscript technically sound, and do the data support the conclusions?

Reviewer #1: Yes

Reviewer #2: Partly

2. Has the statistical analysis been performed appropriately and rigorously? 

Reviewer #1: No

Reviewer #2: No

3. Have the authors made all data underlying the findings in their manuscript fully available?

Reviewer #1: Yes

Reviewer #2: Yes

4. Is the manuscript presented in an intelligible fashion and written in standard English?

Reviewer #1: Yes

Reviewer #2: Yes

5. Review Comments to the Author

Reviewer #1: This article describes the association between systemic vascular resistance in patients after MI.

Comments:

1. Can the authors clarify what factors drive SVR? In their model, it may be limited to hypertension and galectin-3, but there must be more literature and many more factors will play a role. At the very least, can they force age and renal function in their models, as these factors have been proven to affect SVR very substantially. The model limitations must be discussed in a separate limitation section.

2. The review of the literature is substandard. Can the authors elaborate a little more on SVR and progressive remodeling? Is reference#5 up to date?

3. In general, the referencing on galectin-3 is very skewed, key references on the topic are missed, and there is an enrichment towards the authors' own work. It made me uncomfortable. At the very least discuss what is known on galectin-3 in post-MI remodeling.

Reviewer #2: This paper, despite being described as a clinical trial appears to be an assessment of factors in an already extant cohort of patients treated for MI. The abstract would benefit from there being clarity in the setting; at present we are unaware of the fact that n=121 is not directly planned as fulfilling a power calculation with a primary hypothesis and outcome measure, but is in effect a convenience sample where there is no primary outcome or hypothesis given in the design section - it is fundamentally an exploratory and hypthesis generating study. This is not a bad thing - but it needs to be presented as such

It is statistically unsound to dichotomise significance - p=NS must be replaced with actual p-values, and confidence intervals presented. It is unclear what tests are being performed - presumably these are ordinal p-values with one degree of freedom? It is not correct to look at pairwise p-values between two extreme groups unless the global p-value is significant.

It is unclear whether all p-values given have been adjusted for mul;tiple testing as I would expect to see adjusted p-value or some such.

The clear approach here is to perform ordinal regression to identify the risk factors, or indeed to ump SVR++ and SVR+ together depending on the clinical impact of the different groupings here. What is important is the parsimonious multivariable model to predict SVR+, and also the ability to discriminate, which is entirely missing here - there may be a significant difference but the discriminatory value may still be poor. As it stands the paper feels somewhat incomplete, and doesn't provide the required clinical insight of using the factors here as surrogates for diagnosis.

6. PLOS authors have the option to publish the peer review history of their article (what does this mean?). If published, this will include your full peer review and any attached files.

Reviewer #1: No

Reviewer #2: No

---

## [Author Response · Author response to Decision Letter 0]

21 Mar 2020

Reviewer #1: This article describes the association between systemic vascular resistance in patients after MI.

Comments:

1. Can the authors clarify what factors drive SVR? In their model, it may be limited to hypertension and galectin-3, but there must be more literature and many more factors will play a role. At the very least, can they force age and renal function in their models, as these factors have been proven to affect SVR very substantially. The model limitations must be discussed in a separate limitation section.

Response: We agree that the place of conventional factors known to drive SVR was not sufficiently discussed in the article. We have therefore added the following information in the Discussion section (2nd paragraph on p. 19): "Except for physical activity, the main factors known to drive SVR were tested in the present study –i.e. age, kidney function, cardiac function, drug treatment (in particular ACEI/ARBs and beta-blockers) –none of which were found to provide any additional predictive information with regard to that achieved by the baseline levels of Galectin-3, LV end-diastolic volume and SVR. This could at least be partly explained by the low sample size of the present study population. However, this may also be explained by the particular conditions of the post-MI period where SVR are strongly affected by a cascade of adaptive hemodynamic mechanisms and neurohormonal changes and by the introduction of interfering vasoactive treatments, thereby minimizing the impact of other factors."

Accordingly, we have also added the information that the parameters, selected through the multivariable analysis, remained unchanged when age and/or renal function (estimated by the glomerular filtration rate) were forced into the model (last sentence on p. 12).

A small paragraph on the main study limitations was also added in the discussion section (3rd paragraph on p.19).

2. The review of the literature is substandard. Can the authors elaborate a little more on SVR and progressive remodeling? Is reference#5 up to date?

Response: A general paragraph, explaining the relationship between SVR and arterial remodeling has now been added as suggested (2nd paragraph on p. 16).

Reference # 5 has been replaced by a more recent article – i.e. a general review on the vasodilator-related decrease of SVR in heart failure.

3. In general, the referencing on Galectin-3 is very skewed, key references on the topic are missed, and there is an enrichment towards the authors' own work. It made me uncomfortable. At the very least discuss what is known on galectin-3 in post-MI remodeling.

Response: Our apologies for this skewed referencing. The reference of a very recent review article has been added on the value of Galectin-3 in acute myocardial infarction (ref # 20). A paragraph has also been added in the discussion section on this topic (next-to-last paragraph on p. 17) and another paragraph has been expanded accordingly (2nd paragraph on p. 17).

Reviewer #2: This paper, despite being described as a clinical trial appears to be an assessment of factors in an already extant cohort of patients treated for MI. The abstract would benefit from there being clarity in the setting; at present we are unaware of the fact that n=121 is not directly planned as fulfilling a power calculation with a primary hypothesis and outcome measure, but is in effect a convenience sample where there is no primary outcome or hypothesis given in the design section - it is fundamentally an exploratory and hypthesis generating study. This is not a bad thing - but it needs to be presented as such

Response: We agree with all comments from this reviewer and therefore: 

1) this study is now clearly presented as an exploratory analysis of the “REMI” post-MI cohort in the title, as well as in the introduction section of the abstract (p. 2) and main text (first sentences on pp. 4 and 5) 

2) the need for further confirmation through dedicated prospective studies has been highlighted in the abstract (last sentence on p.2) and main text (next-to-last paragraph and last sentence on p. 19). 

It is statistically unsound to dichotomise significance - p=NS must be replaced with actual p-values, and confidence intervals presented. It is unclear what tests are being performed - presumably these are ordinal p-values with one degree of freedom? It is not correct to look at pairwise p-values between two extreme groups unless the global p-value is significant.

Response: We thank the reviewers for her/his comment and accordingly: 

1) exact p-values are now reported for all analyzed variables in Tables 1, 2 and 3 (pp. 10, 11, 13, 14, 15), 

2) all quantitative variables are now reported with median and interquartile range values instead of confidence intervals (certain variables needed to be log-transformed before being reported as mean ± SD values in the previous version of the manuscript), 

3) statistical tests are now better described in the statistical analysis methods section (pp. 7 and 8) –i.e. for the analyses depicted in Tables 1 and 2, non-parametric Kruskall-Wallis tests were performed for continuous variables and Fisher's exact tests for categorical variables, while Wald tests implemented with ordinal logistic regression procedures were used for analyses depicted in Table 3,

4) pairwise p-values are now given in Tables 1 and 2 (P. 10 and 11) only when the global p-values are significant.

It is unclear whether all p-values given have been adjusted for mul;tiple testing as I would expect to see adjusted p-value or some such.

Response: Accordingly, the p-values obtained in the univariable ordinal logistic regressions were adjusted using the Benjamini–Hochberg procedure (Table 3). This information has now been added in the methods section (last sentence of the next-to-last paragraph on p. 7).

The clear approach here is to perform ordinal regression to identify the risk factors, or indeed to ump SVR++ and SVR+ together depending on the clinical impact of the different groupings here. What is important is the parsimonious multivariable model to predict SVR+, and also the ability to discriminate, which is entirely missing here - there may be a significant difference but the discriminatory value may still be poor. As it stands the paper feels somewhat incomplete, and doesn't provide the required clinical insight of using the factors here as surrogates for diagnosis.

Response: We thank the reviewer for her/his comment and accordingly, the regression analyses have now been changed in the new version of the article. These results are now displayed in Table 3 through univariable and multivariable ordinal logistic regression models with risk factors as explanatory variables and the three SVR categories as outcome, namely SVR- (as reference category), SVR+ and SVR++. The statistical method section was modified accordingly (see the last two paragraphs on p. 7 and first paragraph on p. 8).

With this new multivariate analysis, baseline SVR becomes an additional independent predictor (in addition to the baseline levels of Galectin-3 and of LV end-diastolic volume), and this information has now been added in abstract and text. 

Journal Requirements:

Response: This was done accordingly

2. Thank you for including your ethics statement: All subjects gave signed informed consent to participate. The study protocol complied with the principles of the Declaration of Helsinki, was approved by the local Ethics Committee (CPP agreement n° 2009-A00537-50) and registered on the ClinicalTrials.gov site (NCT01109225).

Response: This was done accordingly

Response: This was done accordingly

4. Thank you for including your dual publication statement. To comply with PLOS ONE submission guidelines can you please upload a copy of the previous publication as a supplementary file. Please be aware that if we locate a dual publication issue we will require copyright holder’s permission to publish under CC-By 4.0.

Response: This was done accordingly

---

## [Decision Letter · Decision Letter 1]

20 Apr 2020

Plasma Galectin-3 predicts deleterious vascular dysfunction affecting post-myocardial infarction patients. An explanatory study.

PONE-D-19-29784R1

Dear Dr. Marie,

We are pleased to inform you that your manuscript has been judged scientifically suitable for publication and will be formally accepted for publication once it complies with all outstanding technical requirements.

With kind regards,

Giuseppe Andò, M.D., Ph.D.

Academic Editor

PLOS ONE

Additional Editor Comments (optional):

Reviewers' comments:

Reviewer's Responses to Questions

**Comments to the Author**

1. If the authors have adequately addressed your comments raised in a previous round of review and you feel that this manuscript is now acceptable for publication, you may indicate that here to bypass the “Comments to the Author” section, enter your conflict of interest statement in the “Confidential to Editor” section, and submit your "Accept" recommendation.

Reviewer #1: All comments have been addressed

Reviewer #2: All comments have been addressed

2. Is the manuscript technically sound, and do the data support the conclusions?

Reviewer #1: Yes

Reviewer #2: (No Response)

3. Has the statistical analysis been performed appropriately and rigorously? 

Reviewer #1: Yes

Reviewer #2: (No Response)

4. Have the authors made all data underlying the findings in their manuscript fully available?

Reviewer #1: Yes

Reviewer #2: (No Response)

5. Is the manuscript presented in an intelligible fashion and written in standard English?

Reviewer #1: Yes

Reviewer #2: (No Response)

6. Review Comments to the Author

Reviewer #1: (No Response)

Reviewer #2: (No Response)

7. PLOS authors have the option to publish the peer review history of their article (what does this mean?). If published, this will include your full peer review and any attached files.

Reviewer #1: No

Reviewer #2: No

---

## [Editor Report · Acceptance letter]

29 Apr 2020

PONE-D-19-29784R1 

Plasma Galectin-3 predicts deleterious vascular dysfunction affecting post-myocardial infarction patients. An explanatory study. 

Dear Dr. Marie:

I am pleased to inform you that your manuscript has been deemed suitable for publication in PLOS ONE. Congratulations! Your manuscript is now with our production department. 

With kind regards,

on behalf of

Dr. Giuseppe Andò 

Academic Editor

PLOS ONE